# MHD Thin Film Flow and Thermal Analysis of Blood with CNTs Nanofluid

**Ali Sulaiman Alsagri [1], Saleem Nasir [2], Taza Gul [3] , Saeed Islam [2], K.S. Nisar [4] , Zahir Shah [2] and Ilyas Khan [5],\***

1   Mechanical Engineering Department, Qassim University, Buraydah 51431, Saudi Arabia; a.alsagri@qu.edu.sa
2   Department of Mathematics, Abdul Wali Khan University Mardan, Mardan 23200, Pakistan;
    saleemnasir85@gmail.com (S.N.); saeedislam@awkum.edu.pk (S.I.); zahir1987@yahoo.com (Z.S.)
3   Department of Mathematics, City University of Science and Information Technology (CUSIT),
    Peshawar 25000, Pakistan; tazagul@cusit.edu.pk
4   Department of Mathematics, College of Arts and Science at Wadi Al-Dawaser,
    Prince Sattam bin Abdulaziz University, Al-Kharj 11991, Saudi Arabia; n.sooppy@psau.edu.sa
5   Faculty of Mathematics and Statistics, Ton Duc Thang University, Ho Chi Minh 72915, Vietnam
\*   Correspondence: ilyaskhan@tdt.edu.vn

**Abstract:** Our main objective in the present work is to elaborate the characteristics of heat transport and magneto-hydrodynamics (MHD) finite film flow of human blood with Carbon Nanotubes (CNTs) nanofluids over a stretchable upright cylinder. Two kinds of CNTs nanoparticles, namely (i) SWCNTs (single walled carbon nanotubes) and (ii) MWCNTs (multi walled carbon nanotubes), are used with human blood as a base liquid. In addition, a uniform magnetic field (*B*) has been conducted perpendicularly to the motion of nanoliquid. The transformation of the partial differential structure into a non-linear ordinary differential structure is made by using appropriate dimensionless quantities. The controlling approach of the Homotopy analysis method (HAM) has been executed for the result of the velocity and temperature. The thickness of the coating film has been kept variable. The pressure distribution under the variable thickness of the liquid film has been calculated. The impacts of different variables and rate of spray during coating have been graphically plotted. The coefficient of skin friction and Nusselt number have been presented numerically. In addition, it is noticed that the thermal field of a nanoliquid elevates with rising values of $\phi$ and this increase is more in SWCNTs nanofluid than MWCNTs nanofluid.

**Keywords:** thin film casson nanofluid; SWCNTs and MWCNTs; stretching cylinder; MHD; HAM

## 1. Introduction

### 1.1. Literature Review

Nanofluid, characterized by a significant increase in the heat and mass transfer rate compared to conventional engineered fluid (oils, lubricants, water, ethylene glycol, etc.) [1], is found to serve in a number of engineering applications, for instance, the solar energy system [2], fuel-cell industry [3], petroleum engineering [4–6], materials science [7,8], etc. Choi [9] was the first person who introduced the concept of a dilute suspension of nanoparticles with a dimension less than 100 nm (Cu, $TiO_2$, $Al_2O_3$, Ag, Fe) and their oxides in conventional fluids (oils, lubricants, water, ethylene glycol), which enhances the thermal performance of conventional fluids. Recent applications of nanofluids in the biomechanical field, such as cancer therapy, drug delivery and medicines, have produced a lot of interest in the investigation of nanofluid flows and heat transport. In view of these various applications, researchers have focused their attention on nanofluid flows. Ellahi [10] examined

the impact of MHD and thermal viscosity on the flow of non-Newtonian nanoliquid over a tube. Alshomrani and Gul [11] computed the analytical solution of magneto-hydrodynamics thin film spray of water base $Al_2O_3$ and CuO nanofluids on a horizontal stretchable cylinder. Asadi et al. [12,13] investigated the experimental and theoretical influence of adding MWCNTs, ZnO nanoparticles, and MgO-MWCNT hybrid nanofluids in thermal oil. Gul et al. [14] discussed the impact of an effective Prandtl number on water and ethylene glycol-based alumina nanofluid spray along a stretching cylinder.

CNTs (carbon nanotubes) have a long cylindrical profile, such as frames of carbon atoms with a diameter ranging from 0.7 nm to 50 nm. Carbon nanotubes have a specific importance in nanotechnology, conductive plastics, hardwater, air purification mechanisms, structural composite materials, sensors, display of flat panels, storage of gas, biosensors, extra-long fibers, and many other areas of science and engineering. The idea of CNTs was first discovered in 1991 by Lijima [15]. Carbon nanotubes are further classified as single wall carbon nanotubes (SWCNTs) and multi wall carbon nanotubes (MWCNTs), depending on the number of concentric layers of rolled graphene sheets. Furthermore, carbon nanotubes are predictable inventive material of the 21st century due to their special morphology; new physicochemical features; and unique thermal, electrical, and mechanical characteristics. Additionally, the existence of carbon chains in carbon nanotubes does not pose any danger to the atmosphere. Keeping the above applications, Haq et al. [16] investigated the impact of the thermal conductivity and viscosity of CNTs nanoparticles within three different base fluids (water, engine oil, and ethylene glycol) in nanofluid flowing over a stretching surface. Khan et al. [17] considered the analysis of flow and heat transport of nanofluids containing carbon nanotubes along a flat plate in the presence of the Navier slip boundary condition. Aman et al. [18] examined the effect of MHD on the flow of non-Newtonian CNTs nanofluid. They used three kinds of base liquid. Similarly, the exact solution of Maxwell nanofluid containing CNTs in four types of base fluid was investigated in [19]. Asadi et al. [20–23] conducted some experimental study on the dynamic viscosity of different nanofluids. They found that the viscosity of MWCNTs nanofluids is considerably higher than that of the base fluids. Various other important studies that have been conducted on CNTs base nanofluid can be seen via [24,25].

After the development of nanoliquids, scholars and engineers focused their concentration on examining the motion of nanofluids from various circumstances, such as stretching cylinders and sheets, rotating disks or cylinders, and parallel plates with various flow conditions. The flow problem of magneto-nanofluid through a stretching/extending surface has several practical applications in manufacturing progressions due to the mechanical property of electrically conducting liquids. A stretching surface has gained the extensive attention of scholars due to many manufacturing and technological applications, such as the fabrication and removal of polymer slips from dye, freezing of continuing filaments, lead crystal blowing, manufacture of paper, production of meals, and sketching of wires. Khan et al. [26] explored the phenomena of MHD spray scattering on a stretching cylinder using nanoparticles $Al_2O_3$ and CuO water-based nanoliquids. Recently, some more useful explorations of the subject associated with thin film flow have been presented in [27–30].

Non-Newtonian liquids like toothpaste, food stuff, and plastic have various uses in biochemical, pharmacological, and cosmetic industries. It is very problematic to handle this kind of liquid because the extra nonlinear term originates in the equation of motion. Thus, various liquid models are presented to describe the performance of the said materials. In the present analysis, we select the Casson model. Initially, this model was presented by [31]. It is a shear thinning fluid which is thought of as the zero-shear rate of immeasurable viscosity [32], but is the infinite shear rate at zero viscosity. Human blood, honey, jelly, and soup are examples of Casson fluid. The influence of thermal radiation on Casson fluid flow and the rate of heat exchange on a permeable extending surface have been reported [33]. Asma et al. [34] have explored the MHD flow of Casson liquid on a permeable upright plate. The impact of MHD on Casson nanofliquid flow with thermal radiation over a cylinder was studied in [35].

Among the various models of non-Newtonian time independent fluids models, one of the distinct features and a quite famous Casson model [31,36] known as the most approved is the rheological model for characterizing human blood flow [24,37,38].

In the present work, the thin film Casson nanofluid (human blood) flow comprising CNTs nanoparticles is analyzed with uniform MHD over a stretching upright cylinder. Human blood is used as the base liquid, with two varieties of SWCNTs and MWCNTs nanoparticles inside. The HAM technique [11,39–43] is performed to find the series solution of velocity and the thermal field. The physical performance of each model's parameters for SWCNTs and MWCNTs nanoparticles is presented graphically for velocity, temperature, and pressure fields. Conclusions have been established on the basis of the results.

*1.2. Models of Thermophysical Properties of CNTs Nanofluids*

This section demonstrates the thermophysical properties of CNTs nanofluids.

1.2.1. The Effective Density Model

$\rho_{nf}$ is the effective density of nanofluids, which is given by [24,25]

$$\rho_{nf} = (1 - \varphi)\rho_f + \varphi\,\rho_{CNT}, \tag{1}$$

Here, $\varphi$ signifies the volume fraction of nanoparticles and $\rho_f, \rho_{CNT}$ signify the density of the base fluid and CNTs, respectively.

1.2.2. The Effective Viscosity Model

$\mu_{nf}$ is the effective density of nanofluids, which is given by [24,25]

$$\mu_{nf} = \mu_f(1 - \varphi)^{-2.5}, \tag{2}$$

1.2.3. The Effective Thermal Expansion Coefficient of Nanoparticles Model

$(\rho\beta^{\otimes})_{nf}$ is the effective thermal expansion coefficient of nanoparticles, which is given by [24,25]

$$(\rho\beta^{\otimes})_{nf} = (1 - \varphi)(\rho\beta^{\otimes})_f + \varphi(\rho\beta^{\otimes})_{CNT}, \tag{3}$$

$(\rho\beta^{\otimes})_f, (\rho\beta^{\otimes})_{CNT}$ signify the thermal expansion coefficient of the base fluid and CNTs, respectively.

1.2.4. The Effective Specific Heat Capacity Model

$(\rho\,c_p)_{nf}$ is the effective specific heat capacity of nanofluids, which is given by [24,25]

$$(\rho\,c_p)_{nf} = (\rho\,c_p)_f \left[ (1 - \varphi) + \varphi\left( \frac{(\rho\,c_p)_{CNT}}{(\rho\,cp)_f} \right) \right], \tag{4}$$

$(\rho\,cp)_f, (\rho\,cp)_{CNT}$ signify the specific heat capacity of the base fluid and CNTs, respectively.

1.2.5. The Effective Electrical Conductivity Model

$\sigma_{nf}$ is the effective electrical conductivity of nanofluids, which is given by

$$\sigma_{nf} = \sigma_f \left[ 1 + \frac{3\left( \frac{\sigma_{nf}}{\sigma_f} - 1 \right)\varphi}{\left( \frac{\sigma_{nf}}{\sigma_f} + 2 \right) - \left( \frac{\sigma_{nf}}{\sigma_f} - 1 \right)\varphi} \right], \tag{5}$$

$\sigma_f, \sigma_{CNT}$ signify the electrical conductivity of the base fluid and CNTs, respectively.

### 1.2.6. The Effective Thermal Conductivity Model

In the literature, there are several theoretical models available to calculate the thermal conductivities of carbon nanotubes (e.g., Maxwell's, Jeffery's, Davis's, Hamilton's, and crosser models), but only Xue's model [44] employs principal models, which are effective for spherical and elliptical shape particles. Xue's model was established from the Maxwell model of turning elliptical carbon nanotubes through a big axial ratio and paying the effect of the space sharing on CNTs. Here, for the thermal conductivity of nanofluid $k_{nf}$, Xue's model [44] has been utilized.

$$\frac{k_{nf}}{k_f} = \frac{1 - \varphi + 2\varphi \left( \frac{k_{CNT}}{k_{CNT} - k_f} \right) \ln \left( \frac{k_{CNT} + k_f}{2 k_f} \right)}{1 - \varphi + 2\varphi \left( \frac{k_f}{k_{CNT} - k_f} \right) \ln \left( \frac{k_{CNT} + k_f}{2 k_f} \right)}. \tag{6}$$

$k_f, k_{CNT}$ signify the thermal conductivity of the base fluid and CNTs, respectively.

## 2. Description of Problem

We consider steady and incompressible two-dimensional thin film Casson nanofluids flow along a stretching upright cylinder of radius $a$. The $z$-axis represents along the surface of the cylinder and the $r$-axis is that taken radially, as shown in Figure 1. The cylinder is supposed to electrically conduct with constant $B$ (magnetic field) of strength $B_0$. Here, $T_w = T_a$ is the surface temperature, while $T_\delta = T_b$ is the free surface temperature of the cylinder. In this scenario, the tube surface is stretching with velocity $W_w = 2s\,z$ along the $z$-axis. Here, $s > 0$ is used for extension of the cylinder surface, while for contraction, $s < 0$ is used. Additionally, the thermal field for the present problem is [11]

$$T = T_b - T_r \left( \frac{c\,z^2}{v_{nf}} \right) \Theta(\eta), \tag{7}$$

where $T_r$ is the reference temperature. Furthermore, the human blood-based nanoliquid comprises two sorts of CNTs (SWCNTs and MWCNTs) [24]. Viscous dissipation and natural convection have been involved in nanofluid flow. The stress tensor of the Casson fluid model [36,37] is implemented as

$$\tau_{mn}^{sf} = 2e_{mn}\mu_a^{df} + 2e_{mn}\frac{p_y}{\sqrt{2\pi_d}}, \text{ where } \pi_d \geq \pi_{cr}, \text{ and}$$
$$\tau_{mn}^{sf} = 2e_{mn}\mu_a^{sf} + 2e_{mn}\frac{p_y}{\sqrt{2\pi_d}}, \text{ where } \pi_d \prec \pi_{cr}. \tag{8}$$

In the above expression, the share stress along $m$th and $n$th components is $\tau_{mn}^{sf}$, the deformation rate is $\pi_d$, deformation rate components $m$th and $n$th are $e_{mn}$, the critical value represented by $\pi_{cr}$ is focused on the non-Newtonian fluid model, $\mu_a^{sf}$ is the plastic dynamic viscosity of Casson fluid, and the produce stress of the fluid is $p_y$.

By applying the order analysis, the suggested boundary film equations of carbon nanotubes fluid are [11]

$$\frac{\partial(ru)}{\partial r} + \frac{\partial(rw)}{\partial z} = 0, \tag{9}$$

$$\rho_{nf} \left[ u \left( \frac{\partial w}{\partial r} \right) + w \left( \frac{\partial w}{\partial z} \right) \right] = \mu_{nf} \left( 1 + \frac{1}{\beta} \right) \left[ \frac{\partial^2 w}{\partial r^2} + \frac{1}{r} \left( \frac{\partial w}{\partial r} \right) \right] + (\rho\beta^\otimes)_{nf}(T - T_b)g - \sigma_{nf}B_0^2 w, \tag{10}$$

$$\rho_{nf} \left[ u \frac{\partial u}{\partial r} + w \frac{\partial u}{\partial z} \right] = -\frac{\partial p}{\partial r} + \mu_{nf} \left( 1 + \frac{1}{\beta} \right) \left( \frac{\partial^2 u}{\partial r^2} + \frac{1}{r} \frac{\partial u}{\partial r} - \frac{u}{r^2} \right), \tag{11}$$

$$(\rho c_p)_{nf} \left( u \frac{\partial T}{\partial r} + w \frac{\partial T}{\partial z} \right) = k_{nf} \left( \frac{\partial^2 T}{\partial r^2} + \frac{1}{r} \frac{\partial T}{\partial r} \right) + \mu_{nf} \left( \frac{\partial w}{\partial r} \right)^2. \tag{12}$$

Here, $r, z$ are the radial and axial coordinates, respectively. Additionally, $u(r,z)$ and $w(r,z)$ are the velocity elements in the $r$ and $z$ directions. $\beta = \frac{\mu_a^{sf}\sqrt{2\pi_{cr}}}{\tau_{mn}}$ is the material parameter (Casson parameter); the local pressure and temperature are specified by $p$ and $T$, respectively; the specific density of the nanofluid is $\rho_{nf}$; the dynamic viscosity of the nanofluid is $\mu_{nf}$; $\beta_{nf}^{\otimes}$ is the thermal expansion coefficient of nanoparticles; the electrical conductivity of the nanofluid is $\sigma_{nf}$; the thermal conductivity of the nanofluid is $k_{nf}$; and the specific heat capacity of the nanofluid is $(\rho c_p)_{nf}$.

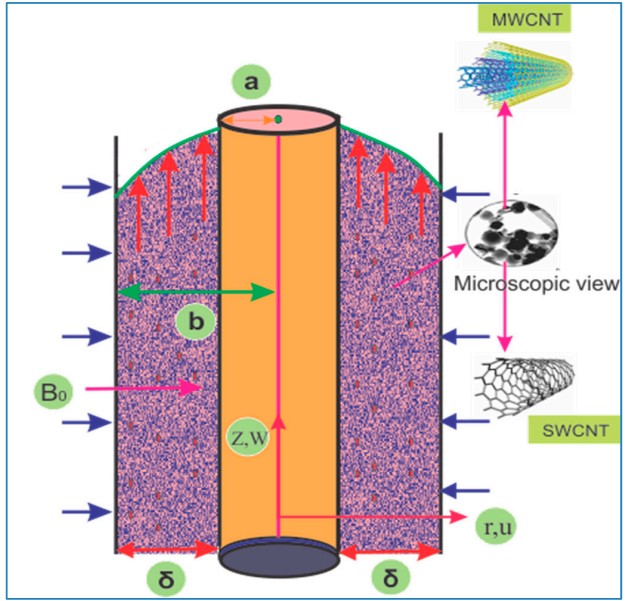

**Figure 1.** Schematic diagram of flow model and coordinate system.

The subjected boundary conditions for the present analysis are as follows [11]:

$$u = U_w, \ w = W_w, \ T = T_w, \ at \ r = a, \tag{13}$$

$$\mu\frac{\partial w}{\partial r} = 0, \ \frac{\partial T}{\partial r} = 0, \ u = w\frac{d\delta}{dz}, \ at \ r = b. \tag{14}$$

where $b$ is the outer radius which display the thickness of the liquid film, the expression of suction and injection velocity is $U_w$, and $W_w$ is the extended velocity of the cylinder surface.

*Non-Dimensional Parameters*

With the aid of the following suitable conversions [11]:

$$\eta = \frac{r^2}{a^2}, \ u = \frac{-sa}{\sqrt{\eta}}[f(\eta)], \ w = 2sz\left[\frac{df(\eta)}{d\eta}\right], \Theta = \frac{T - T_b}{T_a - T_b}. \tag{15}$$

The transformed equations for momentum and energy arise are

$$\left(1 + \frac{1}{\beta}\right)\left[\eta\left(\frac{d^3f(\eta)}{d\eta^3}\right) + \frac{d^2f(\eta)}{d\eta^2}\right] +$$
$$\left((1 - \varphi) + \varphi\frac{\rho_{CNT}}{\rho_f}\right)(1 - \varphi)^{2.5}\left[Re\left(f(\eta)\left(\frac{d^2f(\eta)}{d\eta^2}\right) - \left(\frac{df(\eta)}{d\eta}\right)^2\right) + Gr\Theta(\eta) - Mf(\eta)\frac{df(\eta)}{d\eta}\right] = 0, \tag{16}$$

$$\frac{k_{nf}}{k_f}\left(2\eta\frac{d^2\Theta(\eta)}{d\eta^2} + \frac{d\Theta(\eta)}{d\eta}\right) + PrRe\left((1 - \varphi) + \varphi\frac{(\rho c_p)_{CNT}}{(\rho c_p)_f}\right)\left[f(\eta)\frac{d\Theta(\eta)}{d\eta} - 2\Theta(\eta)\frac{df(\eta)}{d\eta}\right.$$
$$\left. + E_c\left(\frac{d^2f(\eta)}{d\eta^2}\right)^2\right] = 0. \tag{17}$$

The resultant transformed dimensionless boundary conditions are

$$f(\eta) = 1, \frac{df(\eta)}{d\eta} = 1, \Theta(\eta) = 1 \text{ at } \eta = 1, \tag{18}$$

$$\frac{d^2 f(\eta)}{d\eta^2} = 0, \frac{d\Theta(\eta)}{d\eta} = 0, \text{ at } \eta = \alpha. \tag{19}$$

The solid particle volume fraction is $\varphi$, and in non-dimensional form, the variable thickness is

$$\alpha = \frac{b^2}{a^2} = \eta_b. \tag{20}$$

Here, $a, b, \alpha$ represent the radius of the cylinder, the external radius of the thin layer, and the dimensionless thickness of the thin layer, respectively.

$$\text{Re} = \frac{sa^2}{2v_f}, \text{Pr} = \frac{\mu_f(c_p)_f}{k_f}, M = \frac{\sigma_f B_0^2 a^2}{4\mu_f}, Gr = \frac{a^2 g(T - T_0)(\beta \rho)_f}{4W_w \mu_f}, E_c = \frac{W_w^2 a^2}{\Delta T(c_p)_f}. \tag{21}$$

Re is the Local Reynolds number, Pr is the prandtl number, $M$ is the magnetic parameter, $Gr$ is the Grashof number, and $E_c$ is the Eckert number in dimensionless form defined as in [11].

Evaluating the pressure distribution term from Equation (11):

$$\frac{p - p_b}{\mu c_f} = -\frac{Re}{\eta} \left( (1 - \varphi) + \varphi \frac{\rho_{CNT}}{\rho_f} \right) (1 - \varphi)^{2.5} f^2(\eta) - 2 \left( 1 + \frac{1}{\beta} \right) \frac{df(\eta)}{d\eta}. \tag{22}$$

Now, the shear stress at the free surface of the fluid film is zero, which means that

$$\frac{d^2 f(\alpha)}{d\eta^2} = 0. \tag{23}$$

Also, the corresponding shear stress at the cylinder surface is

$$\tau_w = \frac{4s \, z(\rho v)_{nf}}{a} \left( 1 + \frac{1}{\beta} \right) \left[ \frac{d^2 f(1)}{d\eta^2} \right] = \frac{4s \, z\mu_{nf}}{a} \left( 1 + \frac{1}{\beta} \right) \left[ \frac{d^2 f(1)}{d\eta^2} \right]. \tag{24}$$

The non-dimensional forms of $C_f$, $Nu$ (skin friction and Nesselt number, respectively) are expressed as [11]

$$\left[ \frac{zRe}{a} \right] C_f = \left( 1 + \frac{1}{\beta} \right) \left[ \frac{d^2 f(1)}{d\eta^2} \right] (1 - \varphi)^{-2.5}, Nu = -2 \frac{k_{nf}}{k_f} \left[ \frac{d\Theta(1)}{d\eta} \right]. \tag{25}$$

Here, $Re = \frac{sa^2}{2 \, v_f}$ denotes the Reynolds number.

## 3. Solution Methodology

In this paper, we use the HAM technique. The HAM scheme was initially planned by Liao [32,33] and he construed the idea of Homotopy. With the help of HAM, Equations (16) and (17) are solved along with the suggested boundary condition in Equations (18) and (19). To control and improve the convergence of the problem, we used the auxiliary constant $\hbar$. A selection of initial gasses is

$$f_0(\eta) = \frac{\alpha}{2(\alpha - 1)^3} \left[ \eta^3 - 3\alpha\eta^2 - (3 - 6\alpha)\eta + (2 - 3\alpha) \right] + \eta, \Theta_0(\eta) = 1. \tag{26}$$

$L_f$ and $L_\Theta$ are linear operators such that

$$L_f = \frac{d^4 f(\eta)}{d\eta^4} \text{ and } L_\Theta = \frac{d^2 \Theta(\eta)}{d\eta^2}, \tag{27}$$

The general result of $L_f$ and $L_\Theta$ is

$$L_f\left\{K_1 + K_2\eta + K_3\eta^2 + K_4\eta^3\right\} = 0 \text{ and } L_\Theta\{K_5 + K_6\eta\} = 0. \tag{28}$$

For velocity and temperature distribution, the Taylor's expansions have been applied as follows:

$$f(\eta;\rho) = f_0(\eta) + \sum_{\xi=1}^{\infty} f_\xi(\eta)\rho^\xi, \tag{29}$$

$$\Theta(\eta;\rho) = \Theta_0(\eta) + \sum_{\xi=1}^{\infty} \Theta_\xi(\eta)\rho^\xi. \tag{30}$$

But

$$f_\xi(\eta) = \frac{1}{\xi!}\frac{df(\eta;\rho)}{d\eta}\Big|_{\rho=0} \text{ and } \Theta_\xi(\eta) = \frac{1}{\xi!}\frac{d\Theta(\eta;\rho)}{d\eta}\Big|_{\rho=0}. \tag{31}$$

For Equations (16) and (17), the $\xi$th order system is as follows [11]:

$$L_f\left[f_\xi(\eta) - \tilde{N}_\xi f_{\xi-1}(\eta)\right] = \chi_f R_\xi^f(\eta), \tag{32}$$

$$L_\Theta\left[\Theta_\xi(\eta) - \tilde{N}_\xi \Theta_{\xi-1}(\eta)\right] = \chi_\Theta R_\xi^\Theta(\eta). \tag{33}$$

where

$$\tilde{N}_\xi = \begin{cases} 1, & \text{if } \rho > 1 \\ 0, & \text{if } \rho \leq 1 \end{cases}. \tag{34}$$

*Onvergence of HAM*

The HAM scheme has the auxiliary constants $\hbar_f$ and $\hbar_\Theta$ that constantly control and modify the solution convergence. For an appropriate value of $\hbar_f$ and $\hbar_\Theta$, we perform 18th order approximation. The appropriate region $\hbar_f$ and $\hbar_\Theta$ for SWCNTs lies between $-0.1 \leq \hbar_f \leq -0.5$ and $-0.2 \leq \hbar_\Theta \leq 1.9$, while for MWCNTs, the suitable region is between $-0.2 \leq \hbar_f \leq -0.8$ and $-0.1 \leq \hbar_\Theta \leq 1.5$.

## 4. Graphical Results and Discussion

In this portion, we will examine the impact of appropriate model variables on a thin layer flow of Casson nanofluid over a stretching upright cylinder. The main features of the flow, like surface drag force (coefficient of skin friction), the rate of heat transport (Nusselt number), and the rate of spray on the thin layer have been studied for both SWCNTs and MWCNTs nanofluids. The other physical parameters of interest, like the Casson parameter $\beta$, nanoparticles volume fraction $\varphi$, magnetic variable $M$, Grashof number $Gr$, Prandle number Pr, and Reynolds number Re, have been presented graphically and physically discussed for both cases of SWCNTs and MWCNTs nanoparticles. We considered the thin liquid flow and heat analysis of two kinds of CNTs (SWCNTs and MWCNTs) human blood-based nanoliquid. The schematic sketch and coordinate system of the present problem are displayed in Figure 1. The variation in the velocity field $\frac{df(\eta)}{d\eta}$, thermal field $\Theta(\eta)$, and pressure field $\frac{p-p_b}{\mu c_f}(\eta)$ of blood flow against the different emerging parameters of magnitudes ($M = 0.3, \text{Pr} = 24, Ec = 1.5, \varphi = 0.01, \alpha = 1.4, Gr = 0.2, \text{Re} = 0.3$) have been portrayed in the following figures for both sorts of CNTs.

### 4.1. Velocity Distribution

Figure 2 depicts the behavior of $\frac{df(\eta)}{d\eta}$ by varying the magnetic parameter *M* and thin film thickness parameter $\alpha$ of both CNTs (SWCNTs and MWCNTs) nanofluids. The impacts of these quantities on $\frac{df(\eta)}{d\eta}$ are very clear during the flow of both CNTs nanofluids. It can be noted that the larger magnitude of *M* reduced the fluid motion in both cases (SWCNTs and MWCNTs). Physically, such a situation arises as a result of a constantly applied magnetic field $B_0$ that can be induced current in inductive nanoliquid. It creates resistant forces called Lorentz forces, which reduce the liquid velocity. Finally, it is clear that $B_0$ is used to govern the boundary layer separation. Comparatively, a rapid fall in the velocity field is perceived in the case of SWCNTs as related to the MCWNTs. In Figure 2, the effect of $\alpha$ (thin film nanofluid parameter) is depicted for both sorts of CNTs nanofluids. It can be observed that by increasing the value of $\alpha$, the fluid motion decelerates, because in this case, the mass of the fluid is enhanced. Actually, the tiny size of the film accelerates the velocity and less energy is required for fluid motion, for example, the flow in the pipe is much easier and faster than the flow in sea water. Moreover, in the case of MWCNTs, the velocity field is dominant when compared to SWCNTs in the present study.

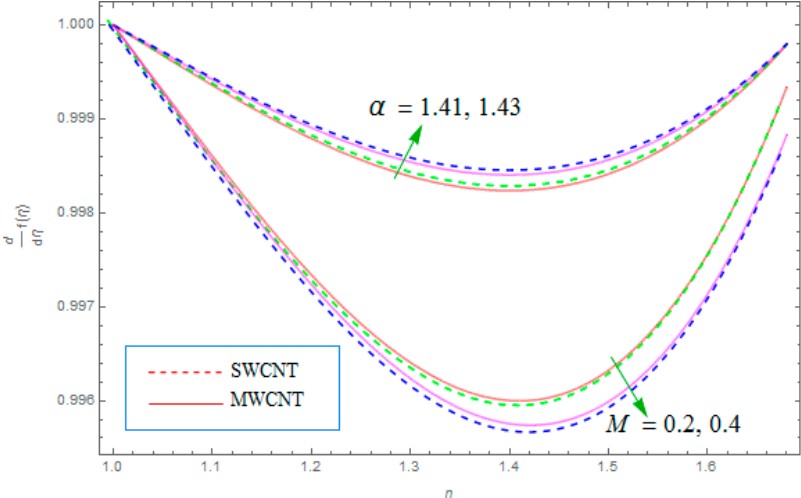

**Figure 2.** $\frac{df(\eta)}{d\eta}$ distribution for varying *M* and $\alpha$.

Figure 3 presents the velocity distribution $\frac{df(\eta)}{d\eta}$ for several values of *Gr* using SWCNTs- and MWCNTs-based nanofluid. The velocity $\frac{df(\eta)}{d\eta}$ elevates for both CNTs by maximizing the value of *Gr*. Similarly, the velocity field $\frac{df(\eta)}{d\eta}$ shows the slowing behavior for both CNTs (SWCNTs and MWCNTs), reducing the value of *Gr*. Actually, the ratio of the thermal buoyancy force in the direction of viscous force is termed the Grashoff number *Gr* Therefore, the basic reason for this is that in the absenteeism of buoyancy force, there is no motion of fluid. The present outline indicates that motion of liquid is occurring due to the buoyancy force and the liquid is stationary in the absence of this force. In addition, it is clear from the figure that SWCNTs are more dominant than MWCNTs.

Figure 4 elucidates the behavior of Casson parameter $\beta$ and nanoparticle volume friction $\varphi$ on $\frac{df(\eta)}{d\eta}$ for both SWCNTs and MWCNTs nanofluids. For the increasing values of $\beta$, the velocity distribution $\frac{df(\eta)}{d\eta}$ in the boundary layer is shown to be declining. It can be noted that accelerating the value of $\beta$ implies condensing the yield stress of Casson liquid and therefore successfully assisting the motion of boundary film flow adjacent to the stretching surface of the cylinder. Furthermore, it is found that the Casson fluid is close to Newtonian fluid for the large value of $\beta \rightarrow \infty$. Similarly, Figure 4 displays the impact of $\varphi$ on the flow of nanoliquid for both nanoparticles (SWCNTs and MWCNTs). Obviously, it is perceived that the velocity distribution $\frac{df(\eta)}{d\eta}$ improves as the magnitude of $\varphi$ increases

for both SWCNTs and MWCNTs nanofluids. Substantially, this occurs by inserting more particles $\varphi$ in the thin fluid of the nanoliquid and increasing the strength of heat carriage and cohesive among the nanoliquid atoms, so that they become frail to halt the faster fluid flow since the heat transport in thin materials is faster than in thick materials. In addition, it is clear from the figure that the flow of SWCNTs is more dominant than MWCNTs. The velocity profiles $\frac{df(\eta)}{d\eta}$ of different magnitudes of the Reynolds number Re for both types of CNTs nanofluids are presented in Figure 5. Basically, Re is the ratio of inertial force toward the viscous force. It can be noted that the velocity profile $\frac{df(\eta)}{d\eta}$ reduces as Re increases, so the velocity tends to be zero at a certain large space from the cylinder surface. Generally, the greater value of Re controlled the inertial force, which reduced the viscous force. Hence, for the larger magnitudes of the Reynolds number Re, the velocity of nanofluids reduces and the flow of fluids declines slowly to the ambient condition. The inertial forces are more influential forces and they do not permit the liquid atoms to flow. Strong viscous forces have a strong resistance to the flow of the liquids. Boundary layer flow of fluid motion decreases with strong inertial forces.

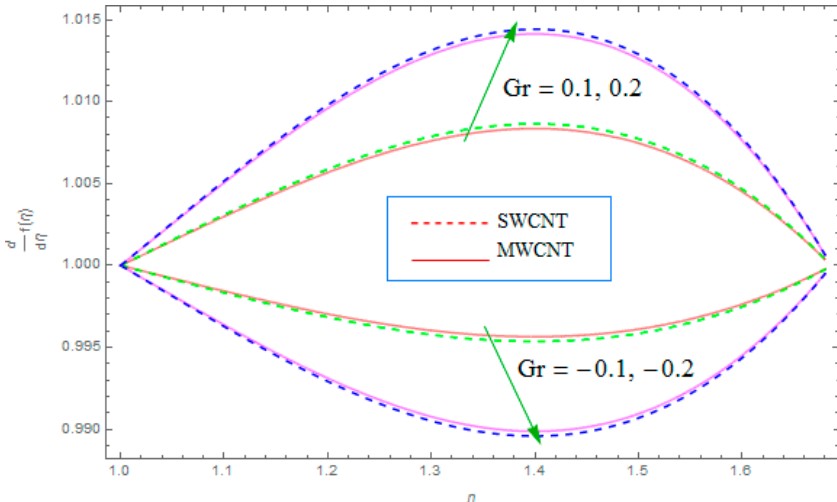

**Figure 3.** $\frac{df(\eta)}{d\eta}$ distribution for varying *Gr*.

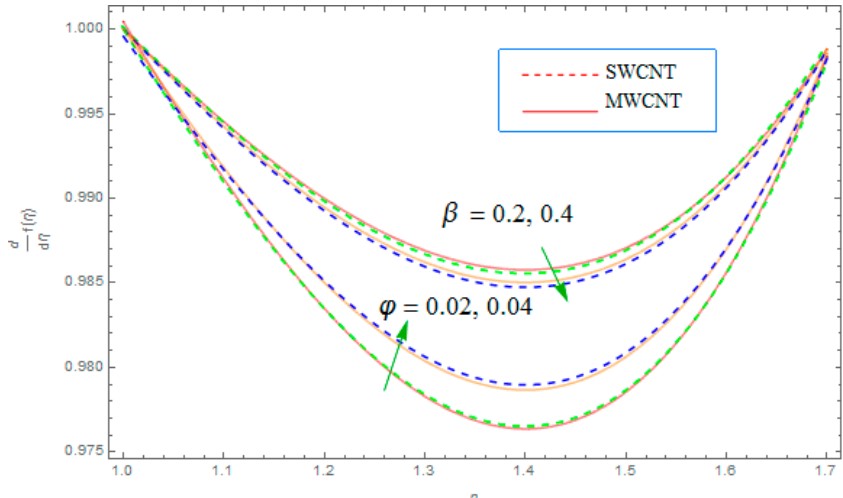

**Figure 4.** $\frac{df(\eta)}{d\eta}$ distribution for varying $\varphi$ and $\beta$.

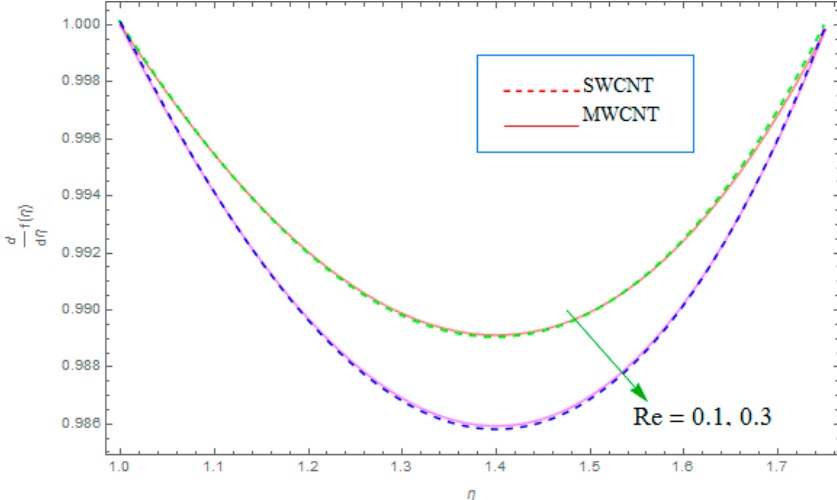

**Figure 5.** $\frac{df(\eta)}{d\eta}$ distribution for varying Re.

### 4.2. Thermal Distribution

Similarly, the next set of Figure 6, Figure 7 displays the impact of flow quantities on the thermal profile $\Theta(\eta)$. Variation in the temperature profile $\Theta(\eta)$ for both types of CNTs nanofluids with magnetic parameter $M$ and thickness variable $\alpha$ is shown in Figure 6. An intensification in the thermal field $\Theta(\eta)$ is perceived with a large value of $M$ for both nanofluids because the high estimation of $M$ produces the Lorentz forces, which increase the fraction force between the fluid molecules for SWCNTs and MWCNTs. This force favors and supports the temperature of fluids. Since the $M$ magnetic field is executed vertically, with the growing magnitude of $M$ magnetic field effect, the fluid is controlled and restricted. Additionally, the greater value of $\alpha$ decreases $\Theta(\eta)$ as the thin liquid coating is heated up quicker than the thick liquid coating. As a result, the thermal field $\Theta(\eta)$ cools down at high values of $\alpha$. The reason for this is that with the thickness of the fluid film, the mass of the fluid increases, which consumes the amount of temperature. Heat enters fluid, and as a result, the environment is cooled down. Thick film fluid needs more heat compared to thin film fluid. Figure 7 demonstrates the performance of the Reynolds number Re and Pr on the thermal filed $\Theta(\eta)$ for SWCNTs and MWCNTs. The same behavior is noted in the variation of Re and Pr for both CNTs. It is seen that a higher measure of Re denigrates $\Theta(\eta)$, explained by the basic fact that a greater magnitude of Re results in extra inertial forces arising, which tightly bonds the particles of flow nanoliquids, and greater heat is enforced to break the contacts amongst the liquid particles. Additionally, the behavior of the Prandtl number Pr on the thermal field $\Theta(\eta)$ is presented in Figure 7. From the figure, it is shown that $\Theta(\eta)$ displays a falling act for a greater magnitude of Pr for both types of CNTs nanoparticles. Generally, a greater magnitude of Pr increases the thickness of the boundary layer, which boosts the cooling efficiency of the nanomaterial. This is because Pr is the ratio of motion diffusivity to thermal diffusivity. Those fluids which have the lowest Prandtl number Pr have good thermal conductivities; therefore, thick boundary layer structures are maintained for diffusing heat.

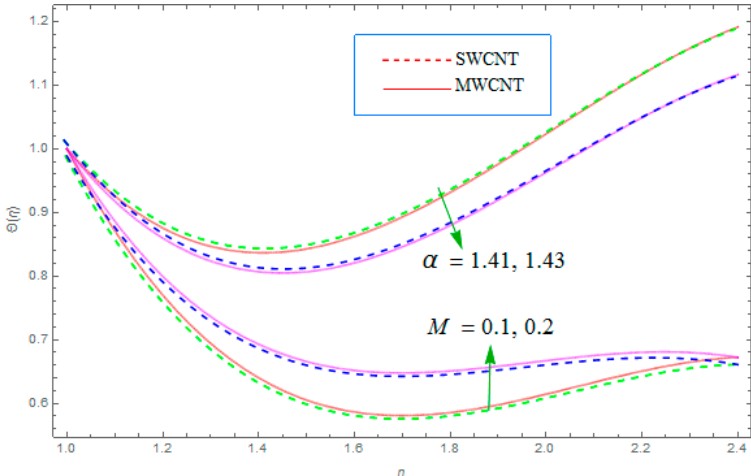

**Figure 6.** Distribution for varying $M$ and $\alpha$.

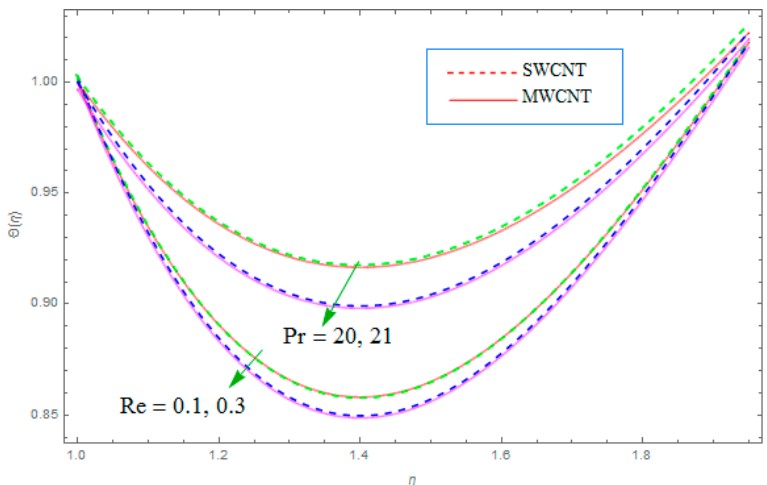

**Figure 7.** $\Theta(\eta)$ distribution for varying Re and Pr.

### 4.3. Pressure Distribution

Finally, in the set of Figures 8 and 9, we portray the variation in the key element of spray phenomena pressure distribution $\frac{p - p_b}{\mu c_f}(\eta)$ in terms of different variables for SWCNTs and MWCNTs nanofluids. The effect of $\varphi$ (volume fraction) and thickness parameter $\alpha$ on pressure distribution $\frac{p - p_b}{\mu c_f}(\eta)$ is sketched in Figure 8. The higher values of $\varphi$ lead to stronger pressure $\frac{p - p_b}{\mu c_f}(\eta)$; as a result, fraction forces are reduced and the concentration of nanoparticles is enhanced for both SWCNTs and MWCNTS nanofluids. Due to a higher concentration, the fluid becomes dense and the collision of molecules increases, exerting pressure on the wall of the cylinder. It has been noticed that the high-pressure phenomena have a vital role in blood flow, chemical reactions, and cooking easily. Furthermore, it can be observed that the pressure distribution $\frac{p - p_b}{\mu c_f}(\eta)$ is enhanced for greater values of $\alpha$. The large size of the film exerts a strong pressure $\frac{p - p_b}{\mu c_f}(\eta)$ and high power is applied to diminish the stress of the thick film. The combined relationship of the film thickness and pressure created a massive force, which is compulsory for the body to move on a fluid surface. Figure 9 exhibits the effect of $M$ and Re on $\frac{p - p_b}{\mu c_f}(\eta)$ for SWCNTs and MWCNTs. From Figure 9, it can be obviously seen that less pressure is produced by a large magnitude of $M$. It can be seen that the pressure distribution $\frac{p - p_b}{\mu c_f}(\eta)$ is weak due to Lorentz forces, which decrease the movement of fluid, and extra pressure is required. The magnetic field $M$ is applied perpendicular to the flow of nanofluids. Therefore, the Lorentz forces capture the liquid in the boundary layer. To compete with the Lorentz forces due to the strong magnetic

field, the pressure must be high in order to cause motion of the fluid. Moreover, large quantities of the Reynolds number Re drop the pressure distribution $\frac{p-p_b}{\mu c_f}(\eta)$ and a strong inertial effect is produced. Due to this inertial force, the fluid particles are packed closely and inflexibly and more pressure is imposed to overcome these forces.

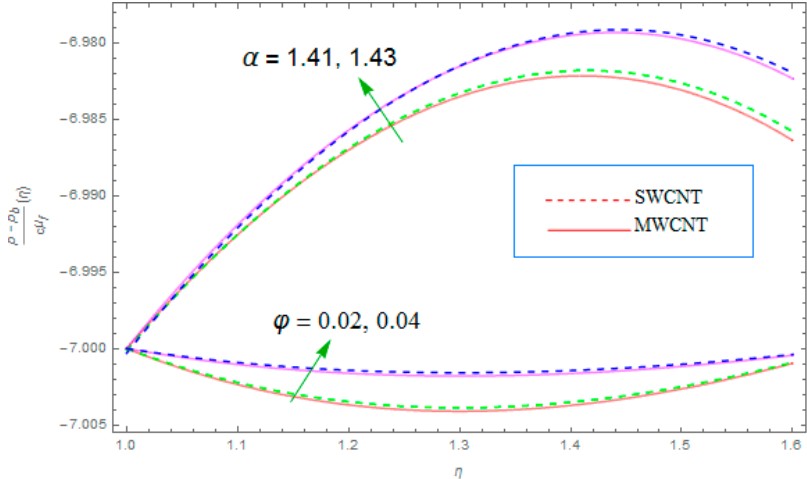

**Figure 8.** $\frac{p-p_b}{\mu c_f}(\eta)$ distribution for varying $\varphi$ and $\alpha$.

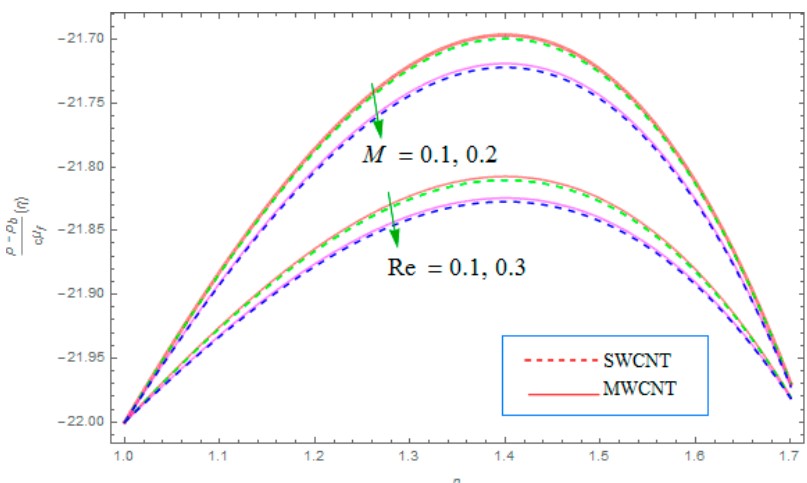

**Figure 9.** $\frac{p-p_b}{\mu c_f}(\eta)$ distribution for varying $M$ and Re.

The certain mathematical values of CNTs (SWCNTs and MWCNTs) and human blood, depend on various thermo-physical characteristics, such as density $\rho$, thermal conductivity $k_f$, and specific heat $c_p$, as presented in Table 1. Also, Table 2 demonstrates the convergence analysis of the series solution for $f''(1)$ (velocity field) and $\Theta'(1)$ (thermal field).

**Table 1.** Various mathematical values of thermophysical characteristics of CNTs of three base liquids [16].

| Physical Properties | | $k$ (W/mK) | $\rho$ (kg/m$^3$) | $c_p$ (J/kgK) | $\beta^{\otimes} \times 10^{-5}$/k | $\sigma$(Sm$^{-1}$) |
|---|---|---|---|---|---|---|
| Base fluid | Human Blood | 0.492 | 1053 | 3594 | 0.18 | 0.8 |
| Nanoparticles | SWCNT | 6600 | 2600 | 425 | 27 | $10^6$–$10^7$ |
| | MWCNT | 3000 | 1600 | 796 | 44 | $1.9 \times 10^{-4}$ |

**Table 2.** The convergence of the Homotopic results for different orders of estimation.

| Order of Approximations | $f''(1)$ | | $-\Theta'(1)$ | |
|---|---|---|---|---|
| | SWCNTs | MWCNTs | SWCNTs | MWCNTs |
| 1 | 0.36550 | 0.31471 | 0.8847 | 0.7056 |
| 5 | 0.32521 | 0.30210 | 0.6667 | 0.5943 |
| 10 | 0.31063 | 0.24323 | 0.6023 | 0.5671 |
| 15 | 0.03117 | 0.02230 | 0.5931 | 0.5395 |
| 18 | 0.02325 | 0.04905 | 0.4299 | 0.4321 |
| 20 | 0.02325 | 0.04905 | 0.3385 | 0.4133 |
| 30 | 0.02325 | 0.04905 | 0.3385 | 0.3133 |
| 37 | 0.02325 | 0.04905 | 0.3385 | 0.3133 |

Table 3 demonstrates the statistical data of $f''(1)$ (surface drag force) HAM approximation at the 20th order for several values of related physical quantities, such as $M = 0.3, \mathrm{Pr} = 24, Ec = 1.5$, $\varphi = 0.01, Gr = 0.2, \alpha = 1.4, \mathrm{Re} = 0.3$ for SWCNTs/MWCNTS nanofluids. In Table 3, it is shown that the magnitude of $f''(1)$ (surface drag force) intensifies for greater values of $\varphi, M, \mathrm{Re}$ for both SWCNTs and MWCNTs. The growing thickness of the nanoparticles enhances the resistance forces, which improve skin friction. The $M$ also governs an opposing force named the Lorentz force and a greater magnitude of $M$ upsurges the skin friction. This drop-in influence is fast using the SWCNTs as compared to the MWCNTs.

**Table 3.** The numerical values of the skin friction coefficient ($f''(1)$).

| $\varphi$ | $M$ | Re | $f''(1)$ | |
|---|---|---|---|---|
| | | | SWCNTs | MWCNTs |
| **0.01** | **0.2** | 0.4 | 0.411834 | 0.508077 |
| 0.02 | | | 0.700187 | 0.968220 |
| 0.03 | | | 0.958088 | 1.273380 |
| 0.01 | 0.3 | | 0.40361 | 0.636512 |
| | 0.4 | | 0.579942 | 0.674852 |
| | 0.3 | 0.5 | 0.551339 | 0.636271 |
| | | 0.6 | 0.579665 | 0.674344 |

Similarly, Table 4 is organized to explain $Nu$ at the 20th order HAM estimate for different values of relevant model variables for both SWCNTs and MWCNTs nanoliquids. From Table 4, it can be clearly observed that the value of rate of heat transport accelerates for a high magnitude of both $\varphi, \mathrm{Pr}$ and declines for a higher value of the $Ec$. The $Ec$ is related to the dissipation term and a larger magnitude of $Ec$ enhances the thermal field. Therefore, the opposite result for the higher magnitude of the $Ec$ verses $Nu$ is perceived.

**Table 4.** The numerical values of the Nusselt number ($-\Theta'(1)$).

| Pr | Ec | $\varphi$ | $-\Theta'(1)$ | |
|---|---|---|---|---|
| | | | SWCNTs | MWCNTs |
| **20** | **1.5** | 0.01 | 0.067258 | 0.201780 |
| 21 | | | 0.107185 | 0.254541 |
| 22 | | | 0.127170 | 0.307476 |
| 20 | 1.6 | | 0.067662 | 0.202577 |
| | 1.7 | | 0.068067 | 0.203375 |
| | 1.5 | 0.02 | 0.246752 | 0.517260 |
| | | 0.03 | 0.427645 | 0.835803 |

## 5. Conclusions

The current study explores the effect of MHD, heat transfer, and pressure distribution of thin layer flow of Casson nanofluid over a stretching upright cylinder. Two forms of CNTs, namely SWCNTs and MWCNTs, were picked as nanoparticles to be applied in human blood base fluid. The obtained set of coupled ODEs was solved by the HAM scheme. The influence of several embedded flow variables on velocity, thermal, and pressure distribution was derived and the derived result was investigated through graphs. The salient features of the current investigation are as follows:

- Increasing the value of the Reynolds number Re and magnetic parameters $M$ yields a reduction in the velocity field for both nanoparticles (SWCNTs and MWCNTs);
- The analysis shows that the volume fraction $\varphi$ increases the velocity, thermal fields, and pressure distribution;
- The important phenomenon of pressure $\frac{p-p_b}{\mu c_f}(\eta)$ declines for a large value of $M$ and Re, while it is enhanced by increasing $\alpha$ and $\varphi$;
- The thermal efficiency of nanofluid improves by increasing the dimension of a nanoparticle, as well as by increasing the magnitude of $\varphi$;
- It is observed that SWCNTs have a greater rate of heat transfer when equated to MWCNTs.

**Author Contributions:** A.S.A. and T.G. modelled the problem and drew the physical sketch. S.N., S.I. and Z.S. introduced the similarity transformation and transformed the modeled problem into dimensionless form. K.S.N. and I.K. solved the problem numerically and computed the results. T.G. and I.K. discussed the results with conclusions. All the authors equally contributed in writing and revising the manuscript.

**Funding:** No specific funding received for this work.

**Conflicts of Interest:** The authors declare no conflict of interest.

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
