# Peer review of "MHD Thin Film Flow and Thermal Analysis of Blood with CNTs Nanofluid"

_coatings, doi:10.3390/coatings9030175_

Round 1
Reviewer 1 Report
The paper deals with the thermal analysis of blood with CNTs nanofluid. It is an exciting piece of research, and my overall recommendation is in favor to publish the paper. However, there are some points which need to be addressed before the publication as follows:
1- The introduction must be strengthened by presenting the most recently published literature on CNT-based nanofluids. It is recommended to present a summary of the recent literature in the form of a table. Followings are some examples of such papers which highly recommended to be included in the literature review:
- A guideline towards easing the decision-making process in selecting an effective nanofluid as a heat transfer fluid.
- Evaluation of MWCNTs-ZnO/5W50 nanolubricant by design of an artificial neural network for predicting viscosity and its optimization.
- An experimental and theoretical investigation on the effects of adding hybrid nanoparticles on heat transfer efficiency and pumping power of an oil-based nanofluid as a coolant fluid.
- Heat transfer efficiency of Al2O3-MWCNT/thermal oil hybrid nanofluid as a cooling fluid in thermal and energy management applications: An experimental and theoretical investigation.
- An experimental and theoretical investigation on heat transfer capability of Mg (OH)2/MWCNT-engine oil hybrid nano-lubricant adopted as a coolant and lubricant fluid.
- The effect of temperature and solid concentration on dynamic viscosity of MWCNT/MgO (20–80)–SAE50 hybrid nano-lubricant and proposing a new correlation: An experimental study.
- Dynamic viscosity of MWCNT/ZnO–engine oil hybrid nanofluid: An experimental investigation and new correlation in different temperatures and solid concentrations.
2- The novelty/novelties should be clearly presented in the last paragraph of the introduction.
3-It is highly recommended to add the core finding of the present research in the abstract.
4- The reasons for using CNTs should be clearly explained in the introduction.
Author Response
Dear Editor,
Thank you for the opportunity to revise our manuscript in accordance with the valuable
comments\ suggestions of reviewers. We incorporate with great care the constructive suggestions of all the reviewers. As a result, the manuscript is substantially improved after making the required changes.
Best Regards
Subject: Reviewer’s comments and Author’s responses.
The paper deals with the thermal analysis of blood with CNTs nanofluid. It is an exciting piece of research, and my overall recommendation is in favor to publish the paper. However, there are some points which need to be addressed before the publication as follows:
Detailed responses of queries and comments of referee first are given below:
Referee’s comment (1)
1- The introduction must be strengthened by presenting the most recently published literature on CNT-based nanofluids. It is recommended to present a summary of the recent literature in the form of a table. Followings are some examples of such papers which highly recommended to be included in the literature review:
- A guideline towards easing the decision-making process in selecting an effective nanofluid as a heat transfer fluid.
- Evaluation of MWCNTs-ZnO/5W50 nanolubricant by design of an artificial neural network for predicting viscosity and its optimization.
- An experimental and theoretical investigation on the effects of adding hybrid nanoparticles on heat transfer efficiency and pumping power of an oil-based nanofluid as a coolant fluid.
- Heat transfer efficiency of Al2O3-MWCNT/thermal oil hybrid nanofluid as a cooling fluid in thermal and energy management applications: An experimental and theoretical investigation.
- An experimental and theoretical investigation on heat transfer capability of Mg (OH)2/MWCNT-engine oil hybrid nano-lubricant adopted as a coolant and lubricant fluid.
- The effect of temperature and solid concentration on dynamic viscosity of MWCNT/MgO (20–80)–SAE50 hybrid nano-lubricant and proposing a new correlation: An experimental study.
- Dynamic viscosity of MWCNT/ZnO–engine oil hybrid nanofluid: An experimental investigation and new correlation in different temperatures and solid concentrations.
Author’s Response (1)
Agreed upon the above comment. The revised article includes the most recently published literature on CNTs-nanofluids. According to comments of Referee’s, the suggested references are added in revised manuscript.
Referee’s comment (2)
The novelty/novelties should be clearly presented in the last paragraph of the introduction.
Author’s Response (2)
In the last paragraph of the introduction, the novelty of the problem is presented.
Referee’s comment (3)
It is highly recommended to add the core finding of the present research in the abstract.
Author’s Response (3)
As suggested, the author adds the main finding in abstract in revised manuscript.
Referee’s comment (4)
The reasons for using CNTs should be clearly explained in the introduction.
Author’s Response (4)
According to the suggested comment of Referee, the author clearly explain the reason of CNTs.
Reviewer 2 Report
It is interesting to investigate the features of MHD thin-film Casson nanofluid flow. Unfortunately, the current form fails to present the study in a coherent and clear manner. Thus, the authors are required to make a major revision by fully according to my following comments before my reconsideration of publication.
1. Fundamental format of the manuscript
Consistence in font, font size, and line space is required for a manuscript, no matter what kind of format is chosen. The authors should check the whole current form, and make necessary revisions, for example Lines 36, 100~101, etc.
In addition, a figure or a table must be presented after it is mentioned for the first time in the text. On a contrast, the authors make most of the figures and tables before they are introduced.
2. Abstract
1) Line 11: “present work” to “the present work”;
2) Line 12: Give the full name of MHD and then use the short name throughout the whole manuscript, as what is done for CNT;
3) Line 15: “forced” to “conducted”;
4) Line 22: I have no idea what the sentence is talking about. It is a highly ill- structured sentence.
5) There are too many (seven) key words, make it no more than five.
3. Introduction
The authors have presented a miserably awful introduction that needs a thorough revision badly. Particularly, the introduction starts in an incorrect way. Neither the concept nor the study of “nanofluid” or “nanoliquid” is “novel”, as the authors stated in Line 30~34. In addition, the authors need to start the introduction by presenting some applications of the nanofluid, implying its importance. For example, they should start it as:
“Nanofluid, characterized by a significant increase in the heat and mass transfer rate compared to conventional engineered fluid [1], is found to serve in a number of engineering applications, for example fuel-cell industry [International Journal of Hydrogen Energy, 2018, 43(37): 17880-17888], petroleum engineering [Fractals, 2018, 26(2):1840015; SPE Journal, 2017, 22(2): 645-659; SPE Production & Operations, 2018, 33(4): 770-783], materials science [Journal of Nanoscience and Nanotechnology, 2017; 17(9):6811-6817; Fractals, 2017, 25(3):1750030], etc.”
Inside the brackets are some necessary recent references to support the statement.
After the start, the authors need to organize the literature review in a coherent way, instead of simply repeating what others have done with details. The literature review should focus on comparing the outcomes of the works, and then evaluating the works naturally. It helps to answer some of the following questions: Why are these works relevant? Which specific problems were addressed? How previous results are related with the proposed work? What are the outstanding, unresolved, research questions? The novelty can then be naturally presented by stating that what specific outstanding issues have been addressed by the proposed study. What others have done and how they have done should be introduced concisely.
On the other hand, the authors should pay attention to the logic that organizes the introduction. For example, the authors stated that they chose Casson model to characterize the liquid in Line 74 and then give a number of literature reviews from Lines 75 to 80. Having done this, the authors provided the reason why to choose Casson model in Lines 81 to 82. They should first give the reasons to choose Casson model, and then review the related studies.
In addition, the English writing is awkward. See the “English writing” part for revision.
4. “Description of the problem” and “Solution methodology”
1) Don’t put an independent equation inside text (for example, Line 96);
2) Since Line 122, the authors presented some theoretical models, which should be moved to “Introduction”;
3) Put Section 2.1 (Dimensionless parameters) into “Solution methodology”, since the involved transformation is a typical method to solve non-linear PDEs;
4) Cite a reference to an equation, unless it is proposed here originally. The authors only did it for some of the equations.
5. Graphical results and discussion
1) Redraw all the figures of results, making the font size does not differ too much in a single figure;.
2) Don’t repeat what the figures have already been illustrated. The authors should focus on “discussion”, the lack of which makes the current form look more like a simple report. The authors should conduct an in-depth discussion and/or analysis of the figures from the perspective of physics, and indicate that why or how some parameters increase or decrease something. This is what interests me most and what can improve the novelty of the manuscript significantly. The authors should keep in mind that they use some existed methods to conduct the study, for example the reduction PDEs to ODEs. If they cannot “dig” more from their results, the novelty will be significantly undermined since the key methods are not new.
3) The English writing is awkward. The authors do need to study how to present the causality by using some written and formal words or phrases, for example, lead to, as a result, etc, instead of writing some cumber sentences, such as “the logic is that” (Line 251).
6. Conclusions
Only the highlights of the key findings are needed in conclusion. The authors presented too much unnecessary information here, including self-evaluation of the novelty of the proposed study (the last item).
7 English writing
The authors have presented a miserably awkward English writing in the current form. They need to make a thorough revision for the improvement, particularly some fundamental mistake. For example:
1) Line 22: highly ill-structured sentences should be revised;
2) Line 30: “have produce” should be “have produced”;
3) Line 31: “which form” should be “formed by”;
4) Line 38: “nanliquids” should be “nanoliquid”. Such a typo implies the authors’ carelessness.
Above are only a few examples of the authors’ awful English writing. They need to check the whole manuscript and then make any necessary revisions.
In a word, major revisions fully according to my comments are required before my reconsideration the manuscript for publication.
Author Response
Dear Editor,
Thank you for the opportunity to revise our manuscript in accordance with the valuable
comments\suggestions of the reviewers. We incorporate with great care the constructive suggestions of all the reviewers. As a result, the manuscript is substantially improved after making the required changes.
Best Regards.
Subject: Referee’s comments and Author’s responses.
It is interesting to investigate the features of MHD thin-film Casson nanofluid flow. Unfortunately, the current form fails to present the study in a coherent and clear manner. Thus, the authors are required to make a major revision by fully according to my following comments before my reconsideration of publication.
Detailed responses of queries and comments of referee’s are given below:
Referee’s comment (1)
Consistence in font, font size, and line space is required for a manuscript, no matter what kind of format is chosen. The authors should check the whole current form, and make necessary revisions, for example Lines 36, 100~101, etc.
In addition, a figure or a table must be presented after it is mentioned for the first time in the text. On a contrast, the authors make most of the figures and tables before they are introduced.
Author’s Response (1)
As suggested, the manuscript has been revised thoroughly and the fonts size and line spacing are corrected. Also, the author’s first explained the figure and tables and then graphically presented.
Referee’s comment (2)
Abstract
1) Line 11: “present work” to “the present work”;
2) Line 12: Give the full name of MHD and then use the short name throughout the whole manuscript, as what is done for CNT;
3) Line 15: “forced” to “conducted”;
4) Line 22: I have no idea what the sentence is talking about. It is a highly ill- structured sentence.
5) There are too many (seven) key words, make it no more than five.
Author’s Response (2)
1- Your comment is completely correct, in line 11 “present work” to “the present work”. It has been corrected in the revised article.
2- As suggested, author add the full name of MHD and CNTs in abstract.
3- In line 15 “forced” to “conducted”; it has been corrected in the revised article.
4- Author’s correct the sentence of line 22.
5- According to the suggestion in revised manuscript, author’s makes five key words.
Referee’s comment (3)
Introduction
The authors have presented a miserably awful introduction that needs a thorough revision badly. Particularly, the introduction starts in an incorrect way. Neither the concept nor the study of “nanofluid” or “nanoliquid” is “novel”, as the authors stated in Line 30~34. In addition, the authors need to start the introduction by presenting some applications of the nanofluid, implying its importance. For example, they should start it as:
“Nanofluid, characterized by a significant increase in the heat and mass transfer rate compared to conventional engineered fluid [1], is found to serve in a number of engineering applications, for example fuel-cell industry [International Journal of Hydrogen Energy, 2018, 43(37): 17880-17888], petroleum engineering [Fractals, 2018, 26(2):1840015; SPE Journal, 2017, 22(2): 645-659; SPE Production & Operations, 2018, 33(4): 770-783], materials science [Journal of Nanoscience and Nanotechnology, 2017; 17(9):6811-6817; Fractals, 2017, 25(3):1750030], etc.”
Inside the brackets are some necessary recent references to support the statement.
After the start, the authors need to organize the literature review in a coherent way, instead of simply repeating what others have done with details. The literature review should focus on comparing the outcomes of the works, and then evaluating the works naturally. It helps to answer some of the following questions: Why are these works relevant? Which specific problems were addressed? How previous results are related with the proposed work? What are the outstanding, unresolved, research questions? The novelty can then be naturally presented by stating that what specific outstanding issues have been addressed by the proposed study. What others have done and how they have done should be introduced concisely.
On the other hand, the authors should pay attention to the logic that organizes the introduction. For example, the authors stated that they chose Casson model to characterize the liquid in Line 74 and then give a number of literature reviews from Lines 75 to 80. Having done this, the authors provided the reason why to choose Casson model in Lines 81 to 82. They should first give the reasons to choose Casson model, and then review the related studies.
In addition, the English writing is awkward. See the “English writing” part for revision.
Author’s Response (3)
The comments of respected reviewer is completely correct, the authors start the introduction by presenting some applications of the nanofluid, implying its importance as suggested by the reviewer. Also, authors add the recommended related references in revised manuscript. Then after start the introduction, the authors organize the literature review in a clear way. Keeping all the above excellent suggestion of reviewer, the author discussed the novelty and outcomes of the present work according to the literature. Furthermore, author add the reason of Casson model in lines 81-82.
Also, the manuscript has been revised carefully and all the grammatical mistakes are corrected to the level best. The typographical mistakes have been removed in the revised manuscript.
Referee’s comment (4)
“Description of the problem” and “Solution methodology”
1) Don’t put an independent equation inside text (for example, Line 96);
2) Since Line 122, the authors presented some theoretical models, which should be moved to “Introduction”;
3) Put Section 2.1 (Dimensionless parameters) into “Solution methodology”, since the involved transformation is a typical method to solve non-linear PDEs;
4) Cite a reference to an equation, unless it is proposed here originally. The authors only did it for some of the equations.
Author’s Response (4)
1- Agreed with the referee suggestion. In line 96 the author makes a separate equation in revised manuscript.
2- As suggestion of the honorable referee, author shifted all the theoretical models in line 122 to introduction.
3- In revised article the section 2.1 put in solution and methodology.
4- All the main equations are cited with a reference in revised manuscript.
Referee’s comment (5)
Graphical results and discussion
1) Redraw all the figures of results, making the font size does not differ too much in a single figure;.
2) Don’t repeat what the figures have already been illustrated. The authors should focus on “discussion”, the lack of which makes the current form look more like a simple report. The authors should conduct an in-depth discussion and/or analysis of the figures from the perspective of physics, and indicate that why or how some parameters increase or decrease something. This is what interests me most and what can improve the novelty of the manuscript significantly. The authors should keep in mind that they use some existed methods to conduct the study, for example the reduction PDEs to ODEs. If they cannot “dig” more from their results, the novelty will be significantly undermined since the key methods are not new.
3) The English writing is awkward. The authors do need to study how to present the causality by using some written and formal words or phrases, for example, lead to, as a result, etc, instead of writing some cumber sentences, such as “the logic is that” (Line 251).
Author’s Response (E)
1- Author redraw all the figures with making the same fonts size.
2- Agreed upon the above comments. The revised article includes some significant physical discussion and novelty of the proposed study and correction/ modification is made where ever needed.
3- The discussion section of manuscript has been revised carefully and all the grammatical mistakes are corrected to the level best. The typographical mistakes have been removed in the revised manuscript.
Referee’s comment (6)
Conclusions
Only the highlights of the key findings are needed in conclusion. The authors presented too much unnecessary information here, including self-evaluation of the novelty of the proposed study (the last item).
Author’s Response (6)
As suggested, the author only presents the key finding and remove all the unnecessary information from the revised manuscript.
Referee’s comment (7)
English writing
The authors have presented a miserably awkward English writing in the current form. They need to make a thorough revision for the improvement, particularly some fundamental mistake. For example:
1) Line 22: highly ill-structured sentences should be revised;
2) Line 30: “have produce” should be “have produced”;
3) Line 31: “which form” should be “formed by”;
4) Line 38: “nanliquids” should be “nanoliquid”. Such a typo implies the authors’ carelessness.
Above are only a few examples of the authors’ awful English writing. They need to check the whole manuscript and then make any necessary revisions.
Author’s Response (7)
The English writing of manuscript has been revised carefully and all the grammatical mistakes are corrected.
1- According to the comments, author has been revised the line 22.
2- Author correct “have produce” should be “have produced”; in line 30 in revised article.
3- Also correct “which form” should be “formed by” in line 31.
4- Correct the typo mistake in revised manuscript.
Whole manuscript has been revised carefully and all the grammatical mistakes are corrected to the level best. The typographical mistakes have been removed in the revised manuscript.
Round 2
Reviewer 2 Report
The revised manuscript is acceptable for publication after some minor revisions:
1) Line 27: “notice” should be “noticed”;
2) The correct format of Refs. 4 and 7 should be:
[4] Xiao, B.; Zhang, X.; Wang, W.; Long, G.; Chen, H.; Kang, H.; Ren, W. A fractal model for water flow through unsaturated porous rocks. Fractals 2018, 26, 2, 1840015.
[8] Xiao, B.; Wang, W.; Fan, J.; Chen, H.; Hu, X.; Zhao, D.; Zhang, X.; Ren, W. Optimization of the fractal-like architecture of porous fibrous materials related to permeability, diffusivity and thermal conductivity. Fractals 2017, 25, 3, 1750030.
3) Capitalize the initial letter of the titles of Refs. 5 and 6, add the missing symbol “-” and the missing “s” such that:
[5] Long, G.; Xu, G. The effects of perforation erosion on practical hydraulic- fracturing applications. 393 SPE Journal 2017, 22, 2, 645-659.
[6] Long, G.; Liu, S.; Xu, G.; Wong, S.; Chen, H.; Xiao, B. A perforation-erosion model for hydraulic-fracturing applications. SPE Production and Operations 2018, 33, 4, 770-783.